# Visualization and functional dissection of coaxial paired SpoIIIE channels across the sporulation septum

Jae Yen Shin[1†], Javier Lopez-Garrido[2†], Sang-Hyuk Lee[1†], Cesar Diaz-Celis[1], Tinya Fleming[2], Carlos Bustamante[1,3,4,5*], Kit Pogliano[2*]

[1]Howard Hughes Medical Institute, University of California, Berkeley, Berkeley, United States; [2]Division of Biological Sciences, University of California, San Diego, La Jolla, United States; [3]Jason L Choy Laboratory of Single Molecule Biophysics, Howard Hughes Medical Institute, University of California, Berkeley, Berkeley, United States; [4]QB3 Institute, University of California, Berkeley, Berkeley, United States; [5]Kavli Energy NanoSciences Institute, University of California, Berkeley, Berkeley, United States

**Abstract** SpoIIIE is a membrane-anchored DNA translocase that localizes to the septal midpoint to mediate chromosome translocation and membrane fission during *Bacillus subtilis* sporulation. Here we use cell-specific protein degradation and quantitative photoactivated localization microscopy in strains with a thick sporulation septum to investigate the architecture and function of the SpoIIIE DNA translocation complex in vivo. We were able to visualize SpoIIIE complexes with approximately equal numbers of molecules in the mother cell and the forespore. Cell-specific protein degradation showed that only the mother cell complex is required to translocate DNA into the forespore, whereas degradation in either cell reverses membrane fission. Our data suggest that SpoIIIE assembles a coaxially paired channel for each chromosome arm comprised of one hexamer in each cell to maintain membrane fission during DNA translocation. We show that SpoIIIE can operate, in principle, as a bi-directional motor that exports DNA.

*For correspondence: carlos@alice.berkeley.edu (CB); kpogliano@ucsd.edu (KP)

†These authors contributed equally to this work

**Competing interests:** The authors declare that no competing interests exist.

## Introduction

The transport of DNA across cellular membranes is an essential part of bacterial processes such as transformation and conjugation (*Errington et al., 2001*; *Burton and Dubnau, 2010*). A paradigmatic example is the segregation of chromosomes that are trapped in the septum during cell division, which requires specialized DNA translocases of the SpoIIIE/FtsK/HerA protein superfamily. The members of this superfamily use the energy of ATP to translocate DNA and peptides through membrane pores (*Bath et al., 2000*; *Iyer et al., 2004*; *Tato et al., 2005*; *Burton and Dubnau, 2010*). SpoIIIE and FtsK contain an N-terminal domain that anchors the protein to the septal membrane (*Wu and Errington, 1997*; *Wang and Lutkenhaus, 1998*; *Yu et al., 1998*), a poorly conserved linker domain, and a cytoplasmic motor domain with ATPase activity that is responsible for DNA translocation. The motdata-left-gapor domain consists of three subdomains: α, β, and γ (*Massey et al., 2006*). α and β form the core ATPase domain and are responsible for chromosome translocation, while the γ subdomain regulates translocation directionality (*Pease et al., 2005*; *Ptacin et al., 2008*).

During *Bacillus subtilis* sporulation, an asymmetrically-positioned septum creates two daughter cells of different size: the bigger mother cell and the smaller forespore. SpoIIIE is made before polar septation (*Foulger and Errington, 1989*) and localizes to the leading edge of the constricting

**eLife digest** *Bacillus subtilis* is a bacterium that lives in the soil and is related to the bacteria that cause the diseases anthrax and botulism in humans. When nutrients are scarce, these bacteria can change into a dormant form called spores, which can withstand harsh environmental conditions. The spores can remain dormant for thousands of years until the conditions improve enough to allow the bacteria to grow again.

During 'sporulation', the membrane that surrounds the bacterium pinches inward near one end of the cell to produce a large mother cell and a smaller forespore. The spore DNA becomes trapped at the site of the division so that the forespore contains only about a third of the DNA of a normal cell. The remaining two thirds lie within the mother cell, and a protein called SpoIIIE is needed to pump this DNA into the forespore. Previous studies have shown that several SpoIIIE proteins team up to form a 'complex' in the membrane that moves the DNA and separates the two cells, but the precise arrangement of SpoIIIE inside cells remained unclear.

Here, Shin, Lopez-Garrido, Lee et al. studied how SpoIIIE is organized in living *B. subtilis* cells, using fluorescent labels to observe the position of SpoIIIE proteins under a microscope. The experiments show that SpoIIIE is arranged as two smaller complexes, one in the mother cell and one in the forespore, each with an equal number of SpoIIIE proteins. This suggests that SpoIIIE assembles into a channel that connects the mother cell and forespore.

To investigate the role of each complex, Shin, Lopez-Garrido, Lee et al. developed a technique called 'cell-specific protein degradation', to destroy SpoIIIE proteins in either the mother cell or the forespore. These experiments show that only the mother SpoIIIE complex is required to move DNA into the forespore, although DNA moves more efficiently when both complexes are present. Furthermore, when SpoIIIE is only present in the forespore, DNA moved out of this cell and into the mother cell. In contrast, both the mother cell and forespore SpoIIIE are required to separate the membranes of the mother cell and forespore.

Shin, Lopez-Garrido, Lee et al.'s findings suggest that SpoIIIE molecules in both cells cooperate to efficiently move DNA into the forespore and to separate the membranes. Further work is required to understand the nature of this cooperation and to determine if similar proteins in other organisms assemble in the same way.

septum (*Fleming et al., 2010*; *Fiche et al., 2013*) (*Figure 1A*). As the sporulation septum closes around the chromosome, SpoIIIE forms a stable focus at the septal midpoint (*Wu and Errington, 1997*; *Fleming et al., 2010*), where it mediates two key events. First, it keeps the mother cell and forespore septal membranes separated in the presence of a septum-trapped chromosome, playing an important role in septal membrane fission (*Liu et al., 2006*; *Fleming et al., 2010*) (*Figure 1*). Second, it translocates the chromosome remaining in the mother cell (about 2/3 of its total length) to the forespore (*Wu and Errington, 1994*; *Bath et al., 2000*). This vectorial DNA translocation is dictated by the interaction of the γ domain with SpoIIIE recognition sequences (SRS) that are distributed in a skewed manner along the *B. subtilis* chromosome from the origin of replication towards the terminus (*Figure 1*) (*Pease et al., 2005*; *Ptacin et al., 2008*). It has been proposed that SpoIIIE exports DNA (*Sharp and Pogliano, 2002*; *Ptacin et al., 2008*) and that the interaction between the γ subdomain of SpoIIIE and the SRS favors either the selective assembly of SpoIIIE in the mother cell or, equivalently, the inactivation or disassembly of motor domains in the forespore (*Sharp and Pogliano, 2002*; *Becker and Pogliano, 2007*; *Ptacin et al., 2008*; *Fiche et al., 2013*).

Structural studies of the motor domains of *B. subtilis* SpoIIIE (*Cattoni et al., 2013*; *2014*) and of *Escherichia coli* and *Pseudomonas aeruginosa* FtsK (*Massey et al., 2006*; *Löwe et al., 2008*) reveal that each assembles a hexamer with a central channel large enough to accommodate one dsDNA molecule. Cell biological studies have shown that both arms of the circular chromosome are translocated simultaneously (*Burton et al., 2007*), suggesting that the translocation complex is made of at least two SpoIIIE hexamers, one for each chromosome arm. However, it remains unclear how SpoIIIE is organized at the septum. One model is that DNA is translocated through an aqueous membrane pore (*Figure 1B*) (*Wu and Errington, 1997*; *Fiche et al., 2013*). According to this model, the

transmembrane domains of SpoIIIE simply localize the complex to the septum, and it has been proposed that tension generated during DNA translocation generates an asymmetric complex with hexameric motor domains oriented towards the mother cell cytoplasm (*Fiche et al., 2013*). A second model proposes that SpoIIIE transmembrane domains in opposite septal membranes pair to form DNA-conducting channel that traverses both septal membranes, with the motor domains in the respective cytosols (*Figure 1C*) (*Liu et al., 2006*; *Burton et al., 2007*; *Fleming et al., 2010*). This model postulates that assembly of the paired channel mediates septal membrane fission and that the assembled channel maintains separation of daughter cell membranes during DNA translocation (*Liu et al., 2006*; *Fleming et al., 2010*).

Here, we investigate the organization of the SpoIIIE DNA translocation complex in living cells. We developed a cell-specific protein degradation system that selectively removes SpoIIIE from each cell after polar septation and used this system with GFP tagging and quantitative PALM (qPALM) to investigate SpoIIIE architecture and function. We show that SpoIIIE forms a complex with approximately equal numbers of molecules in the mother cell and the forespore, enough to assemble at least two hexamers in each cell. Both the forespore and mother cell SpoIIIE subcomplexes are required to maintain septal membrane fission during DNA translocation. However, only the mother cell SpoIIIE is essential for chromosome translocation into the forespore. In the absence of the mother cell protein, forespore SpoIIIE translocates the chromosome out of the forespore, indicating that SpoIIIE exports DNA. Together our results are consistent with the paired channel model in which SpoIIIE assembles a DNA-conducting channel that spans the mother cell and forespore septal membranes. Moreover, we show that SpoIIIE can operate, in principle, as a bi-directional motor that exports DNA from a given cell compartment.

## Results

### Development of a cell-specific protein degradation system

To investigate the organization and the cell-specific function of SpoIIIE we developed a system that allows degradation of specific proteins selectively in the mother cell or forespore during sporulation. Our system is based on a method developed by *Griffith and Grossman (2008)* in which a heterologous SsrA tag from *E. coli* (SsrA*) is fused to the C-terminus of target proteins. The SsrA* tag is recognized by *B. subtilis* ClpXP protease only in the presence of the cognate SspB adaptor protein from *E. coli* (SspB$^{Ec}$). To achieve cell-specific degradation of SsrA*-tagged proteins during sporulation, we expressed *sspB$^{Ec}$* in a cell-specific manner by using promoters (P$_{spoIIQ}$ and P$_{spoIID}$) dependent on the sigma factors σ$^F$ and σ$^E$, which become active only in the forespore or in the mother cell, respectively, immediately after polar septation (*Clarke et al., 1986*; *Rong et al., 1986*; *Londoño-Vallejo et al., 1997*; *Sharp and Pogliano, 2002*) (*Figure 2A*).

We first analyzed the cell-specific degradation of DNA gyrase subunit A (GyrA) by constructing a GyrA-GFP-SsrA* fusion protein. The GFP signal from this protein was initially detected in both the forespore and the mother cell (*Figure 2B*), but disappeared in the SspB$^{Ec}$–containing cell (*Figure 2B*). To estimate the degradation kinetics, we correlated loss of the GFP signal with the stages of engulfment, a phagocytosis like process that occurs during DNA translocation. Immediately after polar septation, the septum is first flat, then curved, and finally the mother cell membrane engulfs the forespore (*Figure 1A*, *Figure 2—figure supplement 1*). Quantification of GFP fluorescence intensity showed that the protein was lost at similar rates in the cell expressing *sspB$^{Ec}$* (forespore or mother cell) (*Figure 2C,D*). In both cases, the fluorescence decreased close to zero when sporangia entered engulfment, indicating that most GyrA-GFP-SsrA* was degraded at this sporulation stage (*Figure 2C,D*). Similar results were obtained after cell-specific degradation of a σ$^A$-GFP-SsrA* fusion protein (*Figure 2—figure supplement 2*). Thus, cell-specific expression of *sspB$^{Ec}$* provides a rapid and efficient way to selectively degrade SsrA*-tagged proteins in the mother cell or in the forespore.

### Cell-specific degradation reveals that the translocation complex contains SpoIIIE molecules in each cell

SpoIIIE translocates DNA across a septum comprised of two membranes, yet it remains unclear if the DNA translocation complex contains SpoIIIE monomers in both the mother cell and the

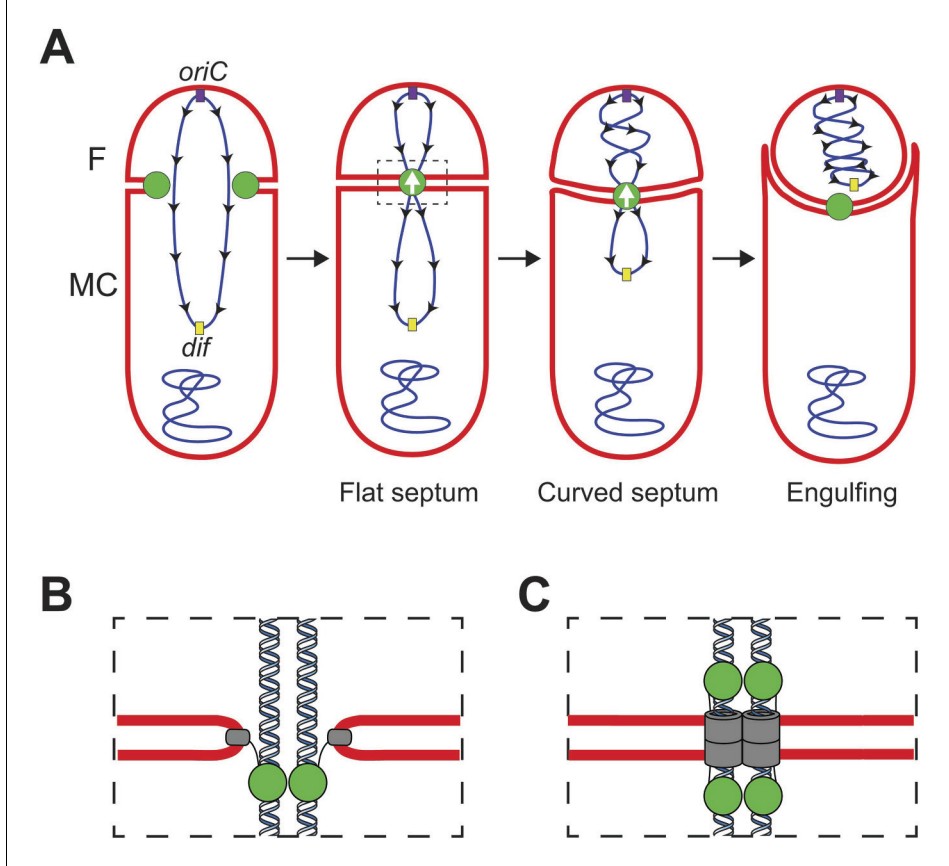

**Figure 1.** Chromosome translocation during *B. subtilis* sporulation. (**A**) The sporulation septum traps the *oriC*-proximal region of the forespore chromosome in the forespore (F), the rest in the mother cell (MC). SpoIIIE (green) localizes at the leading edge of the constricting septum, and assembles a translocation complex at the septal midpoint. The SpoIIIE complex maintains separation of the daughter cell membranes in the presence of trapped DNA. The direction of translocation (white arrow) is determined by the orientation-specific interaction between the SpoIIIE γ domain and the skewed chromosomal recognition sequences known as SRS (black arrowheads on the chromosome, indicating the direction that SpoIIIE motor domains move on the DNA). Engulfment commences during DNA translocation, producing a curved septum and movement of the mother cell membrane around the forespore. (**B**) The aqueous channel model for SpoIIIE, showing two chromosome arms. Green represents the SpoIIIE channels formed by motor domains, grey the transmembrane domains, and red the membrane. (**C**) The paired channel model for SpoIIIE, in which each chromosome arm passes through a proteinaceous channel with subunits in both cells.

forespore or just one cell. To address this question we used the cell-specific degradation system described above. If SpoIIIE assembles in one cell, then triggering degradation in that cell would cause the focus to disappear, while triggering degradation in the other cell would have no effect. However, if the translocation complex is present in both cells, then degradation in only one cell would leave SpoIIIE monomers in the other cell, and the focus would be expected to persist until SpoIIIE was degraded in both cells simultaneously. The SpoIIIE-GFP-SsrA* fusion protein, as expected, formed a bright focus at the septal midpoint and persisted around the forespore during engulfment (*Figure 3A*). When SpoIIIE was degraded in either cell, more than 80% of the sporangia still contained a SpoIIIE focus. However, when SpoIIIE was degraded in both cells simultaneously just 25% of sporangia retained a focus, and these were faint and present only in sporangia with flat and curved septa (*Figure 3A*), which are at early stages of sporulation.

Quantification of focus intensity after degradation in either cell showed that the average focus intensity was reduced by ~50% in sporangia with curved and engulfing septa (*Figure 3B*). After degradation in the forespore, cells with flat septa showed a detectable reduction of GFP intensity

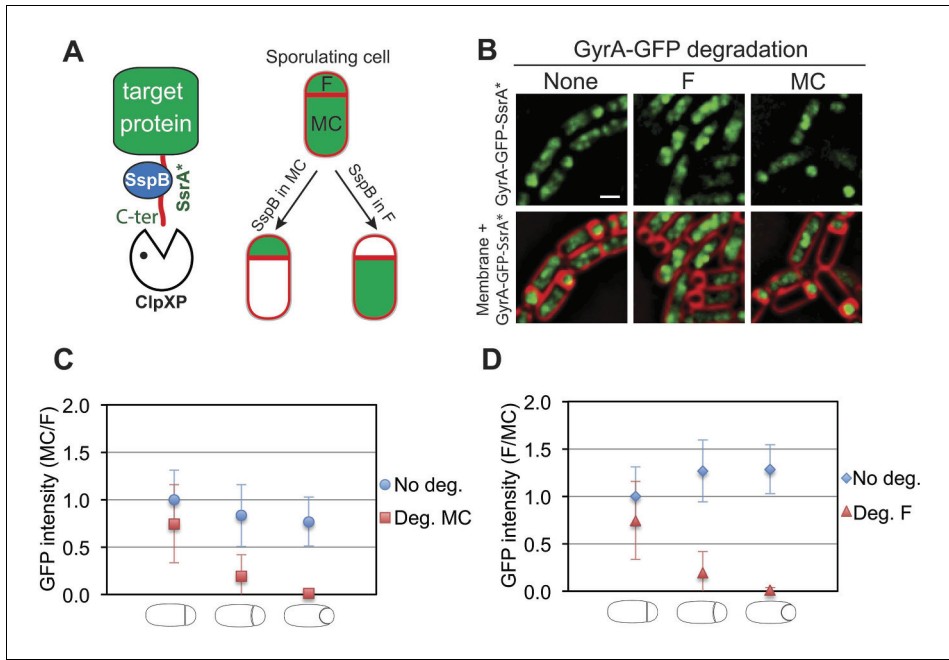

**Figure 2.** Cell-specific degradation of proteins during sporulation. (**A**) Cell-specific protein degradation system (see text). Red indicates the cell membranes, green the target protein. F = forespore, MC = mother cell. (**B**) Fluorescence microscopy of GyrA-GFP-SsrA* (green) during sporulation, without degradation (strain JLG917), forespore degradation (F, JLG919) and mother cell degradation (MC, JLG1281). Membranes are stained with FM4-64 (red). Scale bar, 1 μm. (**C** and **D**) Quantification of the loss of GyrA-GFP-SsrA* fluorescence after degradation in the (**C**) mother cell and (**D**) forespore. No degradation controls express GyrA-GFP-SsrA* but not $sspB^{Ec}$ (blue circles and diamonds). The ratio of the mean GFP intensity in the (**C**) mother cell/forespore (red squares) and (**D**) forespore/mother cell (red triangles) were calculated for sporangia with flat, slightly curved and engulfing septa 2.5 hr after the initiation of sporulation ($t_{2.5}$). 25–66 sporangia were analyzed for each cell type.

The following figure supplements are available for figure 2:

**Figure supplement 1.** Description of the different stages of engulfment analyzed in this study.

**Figure supplement 2.** Cell-specific degradation of σ^A-GFP-SsrA*.

---

(**Figure 3B**), suggesting that SpoIIIE degradation starts slightly faster in the forespore, likely reflecting the earlier activation of σ^F vs σ^E. These results indicate that cell-specific degradation of SpoIIIE commences shortly after polar septation and that the SpoIIIE translocation complex is comprised of monomers with C-terminal motor domains (and the SsrA* tags) in the cytosol of each cell (**Figure 3C**). They also suggest that approximately half of the SpoIIIE molecules in the septal focus are in the mother cell and half in the forespore.

## Direct visualization of SpoIIIE paired channels by PALM

If SpoIIIE assembles a channel with subunits located in each cell, then super-resolution microscopy might show a SpoIIIE cluster in each cell. However, examination of PALM images revealed mainly single foci (**Figure 4A**), consistent with previous reports (**Fleming et al., 2010**; **Fiche et al., 2013**). Most likely this observation reflects the fact that the distance between the forespore and mother cell membranes at the septum is just ~20 nm (**Tocheva et al., 2013**), which is close to the limit of resolution via PALM (~25 nm). Consistent with this idea, occasionally we observed sporangia with what appear to be two clusters of molecules aligned across the septum (**Figure 4A**, right column). To improve the ability to resolve these two subcomplexes, we used a *B. subtilis* mutant that retains thick septal peptidoglycan because it lacks the mother cell transcription factor (σ^E) that controls septal thinning, the first step in engulfment (**Figure 4B**). In this Δσ^E strain, the distance between the mother

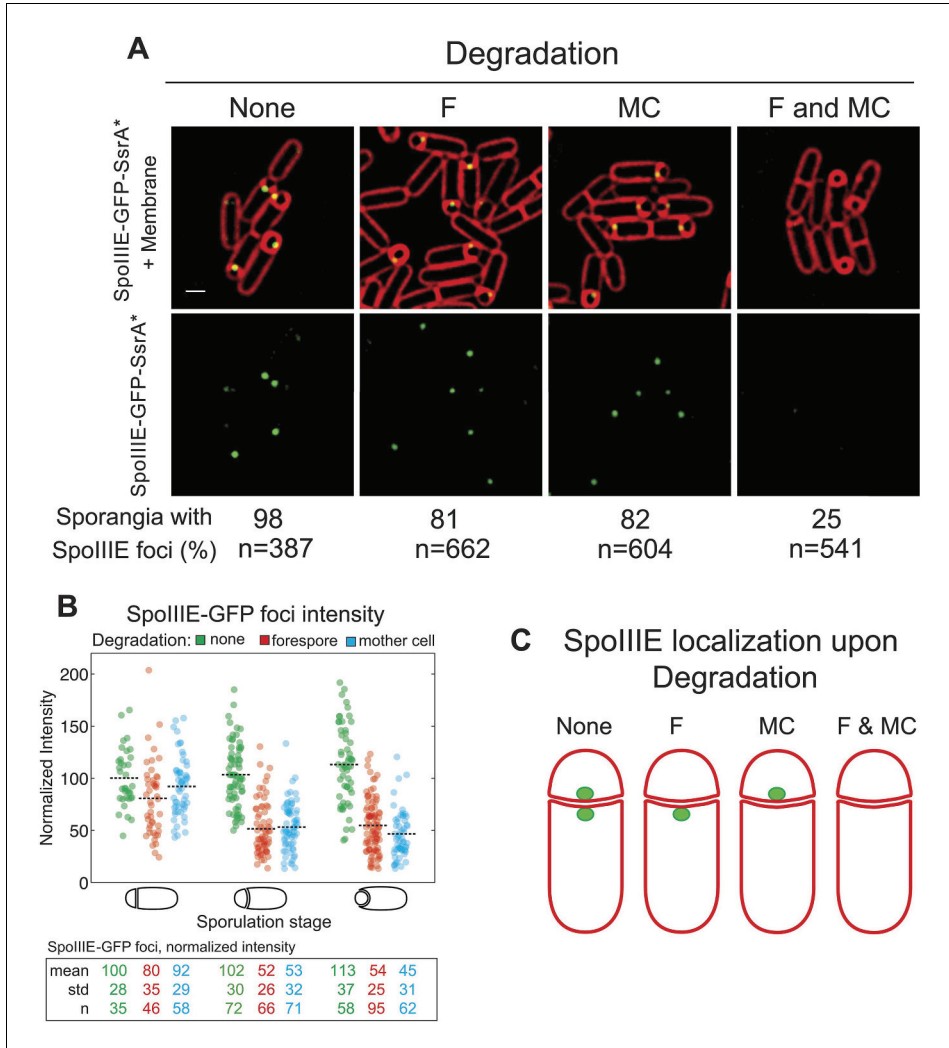

**Figure 3.** Cell-specific SpoIIIE degradation. (**A**) Visualization of SpoIIIE-GFP-SsrA* (green) and FM4-64 stained membranes (red) by deconvolution fluorescence microscopy at $t_{2.5}$. SpoIIIE was not degraded (None, JLG451) or degraded in the forespore (F, JLG452); mother cell, (MC, JLG453) or both (F and MC, JLG454). Percent sporangia with detectable SpoIIIE foci and the number (n) of scored sporangia are indicated for each strain. Strains containing SpoIIIE-GFP-SsrA* without SspB$^{Ec}$ or expressing SspB$^{Ec}$ with untagged-SpoIIIE supported wild-type sporulation, chromosome translocation and membrane fission (Supplemental data). Scale bar, 1 µm. (**B**) Fluorescence intensity of SpoIIIE-GFP foci without degradation (green) or after degradation in the forespore (red) or mother cell (blue) in sporangia with flat, slightly curved and engulfing septa. The mean focus intensity of sporangia with flat septa in the non-degradation strain was normalized to 100. Each dot represents the intensity of a single focus; black dotted lines represent the mean of each data set. Between 35 and 95 foci were analyzed for each data set. (**C**) Model for SpoIIIE organization at the septal midpoint based on cell-specific degradation. The translocation complex contains SpoIIIE molecules on both sides of the septum so cell-specific degradation will only remove a fraction of the molecules. Degradation in both cells will remove all molecules.

cell and forespore septal membranes is ~40 nm, more than twice the distance of wild type (*Illing and Errington, 1991*), potentially allowing resolution of the SpoIIIE subcomplexes in each cell by PALM. We therefore introduced SpoIIIE-tdEOS into the Δσ$^E$ strain. As expected (*Pogliano et al., 1999*), this strain did not initiate engulfment or block the second potential division septum, producing a high number of disporic sporangia (*Figure 4C*, *Figure 4—figure supplement 1*) without impairing DNA translocation (*Figure 4—figure supplement 2*).

Visualization of SpoIIIE-tdEOS via PALM in this strain showed that 39% of septa contained a medial SpoIIIE assembly comprised of two separate subcomplexes of molecules (hereafter called

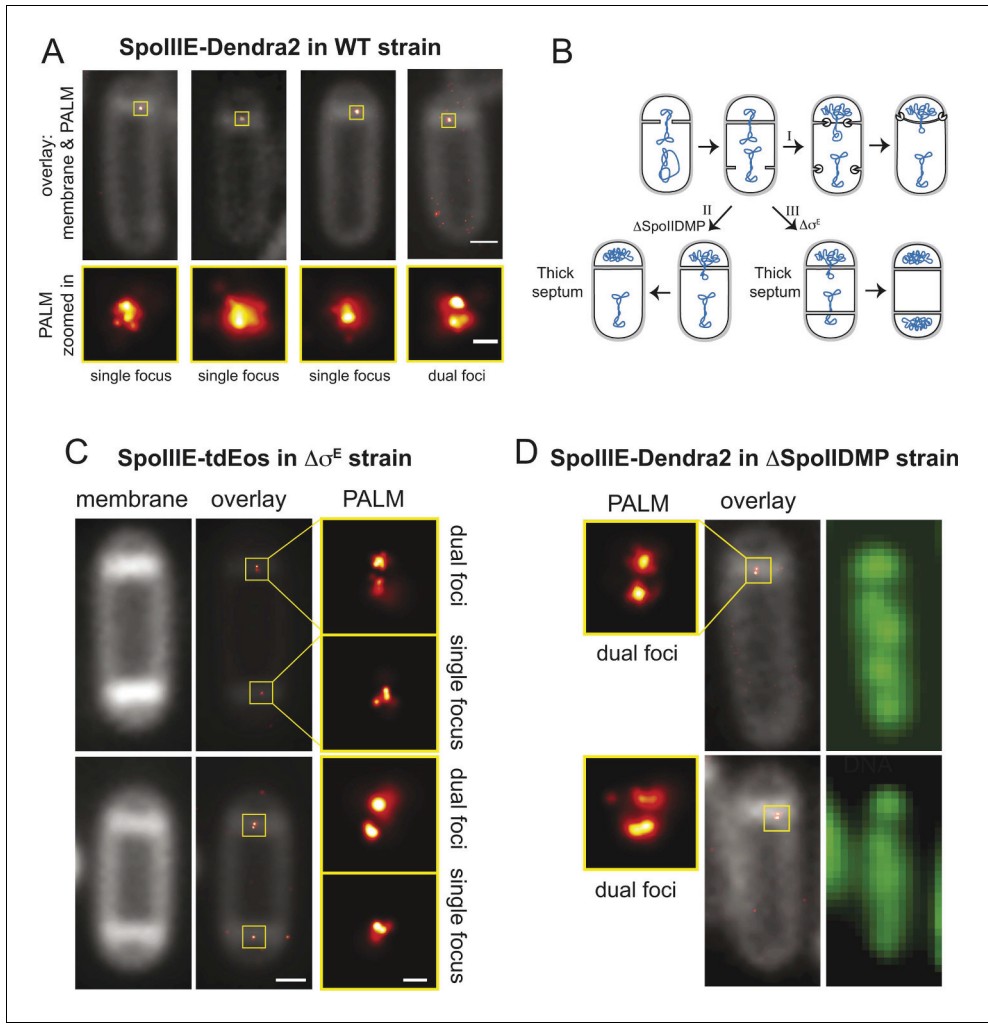

**Figure 4.** Direct visualization of two SpoIIIE clusters at the septal midpoint. (**A**) PALM image of single focus and a rare dual focus of SpoIIIE-Dendra2 in wild type *B. subtilis* (TCF25), with FM5-95 stained membranes (white). The bottom PALM images are zoomed in from the yellow boxes. Bar is 500 nm and 50 nm for overlaid and PALM images, respectively. Membrane is diffraction-limited image. (**B**) Schematic diagram of septal thinning. (I) After septation, septal peptidoglycan is degraded by a complex containing the SpoIID, SpoIIM and SpoIIP proteins (pacman) and the second potential division site is blocked. (II) Elimination of SpoIIDMP in the ΔspoIIDMP (*spoIID*, *spoIIM*, *spoIIP*) strain inhibits septal thinning without impairing DNA segregation. (III) Elimination of σ$^E$ in the Δσ$^E$ (*spoIIGB*) strain inhibits septal thinning and produces disporic cells without impairing DNA segregation. (**C**) PALM images of SpoIIIE-tdEos in Δσ$^E$ strain (JS03). Classification of PALM images as dual foci or single foci were defined according to the parameters of our cluster analysis. Scale bar is 500 nm and 50 nm for overlaid and PALM images, respectively. Membrane is diffraction-limited image. (**D**) PALM images of SpoIIIE-tdEos in ΔSpoIIDMP strain (JLG571). The diffraction-limited images of the membranes (white) and the DNA (green) were obtained by staining with FM5-95 and DAPI, respectively. The relative forespore DAPI intensity was ~25% in both sporangia. SpoIIIE dual foci are shown in the left panels. In-gel fluorescence and western blots of the different SpoIIIE fusion proteins used here can be found in *Figure 4—figure supplement 7*.

The following figure supplements are available for figure 4:

**Figure supplement 1.** Classification of cells in vegetative, monosporic and disporic cells at $t_{1.75}$.

**Figure supplement 2.** SpoIIIE segregates the chromosomes in the Δσ$^E$ strain.

**Figure supplement 3.** Examples of single foci in the Δσ$^E$ strain when SpoIIIE$^{WT}$ is fused to tdEos.

*Figure 4 continued on next page*

*Figure 4 continued*

**Figure supplement 4.** Distribution of single and double foci in monosporic and disporic sporangia.

**Figure supplement 5.** SpoIIIE ATPase mutant (SpoIIIE$^{ATP-}$) fused to tdEos in Δσ$^E$ strain also organizes into dual foci.

**Figure supplement 6.** SpoIIIE organizes into dual foci at the septum of sporangia where chromosome transport is incomplete.

**Figure supplement 7.** In-gel fluorescence and western blot of the SpoIIIE fusion proteins used in this study.

'dual foci', *Figure 4C,D*). Most dual foci were aligned across the division septum and a few were slightly tilted (*Figure 4C*). The average distance between the centers of each focus was 55 nm (*Figure 5A,B*, *Figure 5—figure supplement 1*). Electron cryotomography shows that before septal thinning the cytoplasmic faces of the septum are separated by ∼40 nm (*Tocheva et al., 2013*), suggesting that the centers of each resolved subcomplexes of SpoIIIE lie in separate cells. Dual foci were also observed in 26% of sporangia when septal thinning was blocked by the absence of the SpoIID, SpoIIM and SpoIIP proteins that degrade septal peptidoglycan (*Figure 4B,D*) (*Pogliano et al., 1999*; *Abanes De Mello et al., 2002*; *Chastanet and Losick, 2007*; *Gutierrez et al., 2010*). Thus using two strains with thick septa allowed visualization of SpoIIIE foci comprised of molecules in both the mother cell and forespore (*Figure 4*).

We reasoned that if the 61% of septa that showed single foci in the Δσ$^E$ strain (*Figure 4C*, *Figure 4—figure supplement 3*) were unresolved dual foci, they should be elongated across the septum compared to each focus in septa with dual-foci. We therefore determined the width of the SpoIIIE clusters in single foci and in each resolved subcomplexes from dual foci both perpendicular and parallel to the septum. On average, the width perpendicular to the septum of the single foci (94 nm) was ∼1.7× larger than that of each individually resolved cluster in dual foci (54 nm), while their width parallel to the septum was similar (∼80 nm, *Figure 5C*), suggesting that single foci are unresolved dual foci. Consistent with this hypothesis, PALM images of SpoIIIE-Dendra2 showed a larger (83%) number of septa with a single focus than SpoIIIE-tdEOS (*Figure 4—figure supplement 4*), as expected for a dimmer fluorescent protein with reduced localization accuracy (*Lee et al., 2012*).

## Dual foci are observed when DNA traverses the septum

We performed two experiments to determine if the dual foci represented the architecture of the SpoIIIE complex during or after DNA translocation. First, we examined PALM images of the translocation-deficient SpoIIIE ATPase mutant (G467S; hereafter SpoIIIE$^{ATP-}$) in Δσ$^E$ background. This strain showed dual foci in 23% of septa (*Figure 4—figure supplement 5*), suggesting that dual foci represent the structure of SpoIIIE when it assembles around the trapped chromosome, not after DNA translocation. Second, we used PALM to localize wild type SpoIIIE-tdEOS in ΔSpoIIDMP strain in sporangia stained with both the membrane stain FM5-95 and the DNA stain DAPI, so we could discriminate between cells in which DNA translocation was ongoing and those in which it was complete. Quantification of the forespore DNA relative to the total (mother cell plus forespore), demonstrated that 81% of sporangia with dual foci had not completed DNA translocation (*Figure 4D*, *Figure 4—figure supplement 6*). These data suggest that dual foci represent the SpoIIIE DNA translocation complex.

## The mother cell and forespore subcomplexes have similar numbers of SpoIIIE molecules

We next used qPALM to determine the relative numbers of SpoIIIE molecules in each cell in Δσ$^E$ sporangia with resolved dual foci, applying an algorithm recently developed in our laboratory (*Lee et al., 2012*; see 'Materials and methods' for details). We detected, on average, 34 ± 17 molecules in the entire dual foci, consistent with the number of molecules of wild type SpoIIIE and SpoIIIE$^{ATP-}$ in strains with thin septa (*Figure 5D*, *Table 1*). We detected 19 ± 11 and 15 ± 8 molecules in the mother cell- and forespore-proximal subcomplexes from dual foci, respectively. Thus

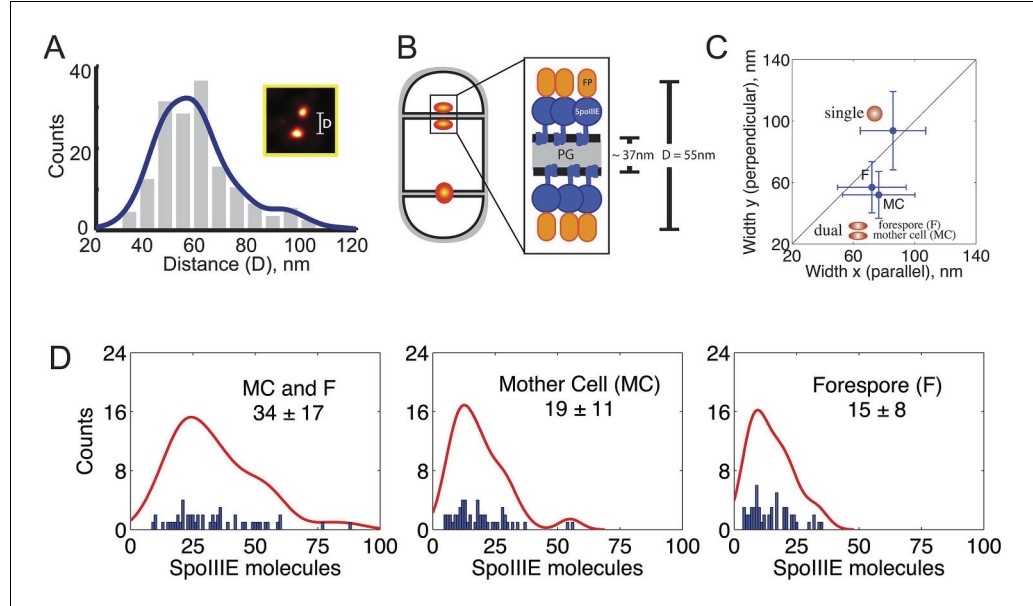

**Figure 5.** quantitative PALM (qPALM) of SpoIIIE complexes. (**A**) The distances (D) between the centers of each cluster from SpoIIIE-tdEos dual foci in $\Delta\sigma^E$ strain (JYS03). The average distance between clusters, indicated by the main peak value of the kernel density estimator (blue line), is 55 nm. (**B**) Schematic diagram of SpoIIIE (blue) at the division septum in $\Delta\sigma^E$ strain. To estimate the thickness of the peptidoglycan (PG) we subtracted twice the width of the lipid bilayer (3 nm; **Lewis and Engelman, 1983**) and twice the height of SpoIIIE (6 nm), based on the FtsK crystal structure (**Massey et al., 2006**) from 55 nm, to give 37 nm. FP, fluorescent protein. (**C**) Dimensions of single SpoIIIE-Dendra2 foci and resolved clusters in dual foci. Focus widths in foci parallel (x) and perpendicular (y) to the septum were calculated as described in supplementary methods. For single foci the parallel- and perpendicular-width (in nm) are 86 ± 10 and 94 ± 12 respectively, for dual foci, the forespore proximal cluster is 64 ± 11 and 57 ± 8, the mother cell proximal cluster 77 ± 12 and 52 ± 7. Error bars, standard deviation from $N_{single}$ = 221, $N_{dual\ foci}$ = 51. (**D**) Distribution of the number of SpoIIIE-Dendra2 molecules determined by qPALM at the septum in the mother cell, in the forespore and in both. Data (blue bars) are represented by Kernel Density Estimator (red solid lines). Mean and standard deviations from $N_{foci}$ = 51 are shown. More details about qPALM and the algorithm employed for quantification can be found in 'Materials and methods'.

The following figure supplements are available for figure 5:

**Figure supplement 1.** Distance distribution between the centers of each of the clusters from the SpoIIIE-Dendra2 dual foci in $\Delta\sigma^E$ strain.

**Figure supplement 2.** Fermi-photoactivation.

each subcomplex contains approximately half the number of SpoIIIE molecules detected in the entire dual focus, consistent with our quantification of the fluorescence intensity after cell-specific degradation of SpoIIIE-GFP (*Figure 3B*, *Table 1*). Together these results suggest that the SpoIIIE translocation complex consist of two complexes containing equivalent numbers of SpoIIIE molecules, one in the mother cell and one in the forespore.

## Septal membrane fission is reversible and requires SpoIIIE in both cells

The last step of cell division is a membrane fission event that divides the septal membrane to render two physically separated daughter cells. During sporulation, septal membrane fission occurs in the presence of a trapped chromosome (*Burton et al., 2007*; *Fleming et al., 2010*) and depends on the assembly of a stable SpoIIIE translocation complex (*Fleming et al., 2010*). It has been proposed that SpoIIIE mediates membrane fission by assembling a paired channel spanning both septal membranes (*Liu et al., 2006*; *Fleming et al., 2010*), with both halves of the channel required to mediate and maintain septal membrane fission. To test this hypothesis, we used cell-specific SpoIIIE

degradation together with fluorescence recovery after photobleaching (FRAP) to assess the continuity of the forespore and mother cell membranes after degrading SpoIIIE in one or the other cell. Briefly, sporangia were stained with the membrane dye FM4-64 and the dye in the forespore photobleached with a laser. If the membrane of the mother cell and the forespore are connected to form a continuous unit, FM4-64 will diffuse from the mother cell to the forespore and fluorescence will completely recover (**Figure 6A**, right). However, if the membranes are separated and physically disjointed, FM4-64 will not diffuse and forespore fluorescence will not recover (**Figure 6A**, left).

Since SpoIIIE is only essential for septal membrane fission in the presence of trapped DNA, we used SpoIIIE$^{ATP-}$, which supports septal membrane fission (**Fleming et al., 2010**) but not chromosome translocation (**Sharp and Pogliano, 1999**). We detected the same number of molecules in the foci of SpoIIIE$^{ATP-}$ as the wild type (**Table 1**), suggesting that the organization of the complex is similar. We constructed a GFP-SsrA* tagged version of SpoIIIE$^{ATP-}$, which was degraded similarly to the wild-type allele (**Figure 6—figure supplement 1**), and performed FRAP of engulfing sporangia. Fluorescence did not recover when SpoIIIE was not degraded, indicating that the membranes are separated (**Figure 6B,C**, **Figure 6—figure supplement 2**). However, fluorescence completely recovered in most sporangia within 60 s when SpoIIIE was degraded in the mother cell, the forespore or both (**Figure 6B,C**, **Figure 6—figure supplement 2**). These results indicate that SpoIIIE subcomplexes in both cells are required for septal membrane fission, supporting the model that assembly of a paired channel contributes to membrane fission (**Figure 7C**). Furthermore, the observation that the separated membranes before degradation were converted to continuous membranes after degradation suggests that membrane fission is reversible during DNA translocation and depends on the integrity of the paired channel.

## DNA translocation involves different contributions by forespore and mother cell SpoIIIE

We next used the cell-specific degradation system to determine if both SpoIIIE subcomplexes were necessary for chromosome translocation, taking advantage of the observation that SpoIIIE degradation occurs before chromosome translocation is complete (**Figure 7—figure supplement 1**). We used the SpoIIIE-GFP-SsrA* cell-specific degradation strains and quantified the amount of DAPI-stained DNA in the forespore relative to the total DNA in sporangia that were about to complete engulfment (**Figure 7A,B**). As previously reported (**Becker and Pogliano, 2007**; **Ptacin et al., 2008**), when chromosome translocation is completed the normalized forespore DAPI intensity is just ~30–40% (of the total DNA) rather than the expected 50% (one full chromosome in the forespore and the other in the mother cell), likely due to self-quenching of DAPI in the smaller volume of the forespore. In the SpoIIIE$^{ATP-}$ mutant, where chromosome translocation is blocked, and before DNA translocation starts in the wild type, the normalized forespore DAPI intensity is 10–15% (**Figure 7—figure supplement 1**). Without degradation, most sporangia finished chromosome translocation before the completion of engulfment showing an average normalized forespore DAPI intensity of 37% (**Figure 7B**). Similarly, when SpoIIIE was degraded in the forespore, most sporangia completed chromosome translocation (normalized DAPI intensity, 38%). However, 26% of sporangia showed a normalized DAPI intensity <30% (**Figure 7B**), suggesting that in some sporangia the absence of the forespore complex impaired the ability of the mother cell complex to translocate DNA. In contrast, degradation of SpoIIIE in the mother cell blocked chromosome translocation (normalized DAPI intensity, 20%) in most sporangia and 20% of sporangia showed anucleate forespores (**Figure 7A,B**), indicating that in the absence of the SpoIIIE mother cell subcomplex, the forespore subcomplex can

**Table 1.** Summary of SpoIIIE quantification obtained by qPALM and diffraction-limited images in the wild type strain, cell-specific degradation system and the thick septum strain (Δσ$^E$)

| Strain | SpoIIIE-dendra2 quantification by qPALM, absolute numbers | | | | | SpoIIIE-GFP intensity, % | | |
| | Wild type | | Δσ$^E$ (dual foci) | | | Cell-specific degradation | | |
| Strain | SpoIIIE$^{WT}$ | SpoIIIE$^{ATP-}$ | MC + F | MC | F | None | F | MC |
| Mean (SD) | 31 (18) | 31 (14) | 34 (17) | 19 (11) | 15 (8) | 100 (29) | 51 (25) | 52 (31) |
| N | 91 | 128 | 51 | 51 | 51 | 72 | 66 | 123 |

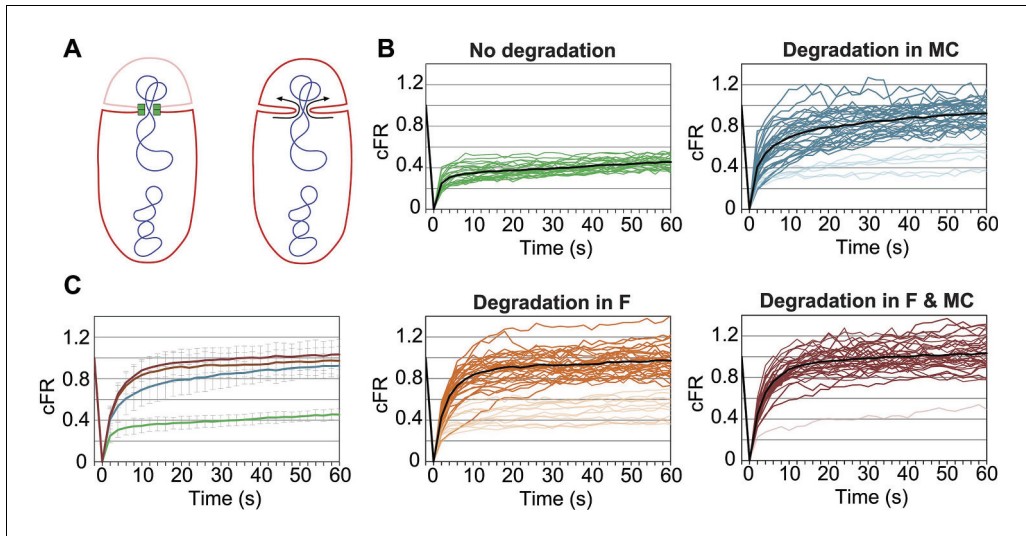

**Figure 6.** Role of forespore and mother cell SpoIIIE in membrane fission. (**A**) Membrane organization in wild type (left) and ΔSpoIIIE (right) strains. SpoIIIE (green) blocks the FM4-64 (red) diffusion from the mother cell to the forespore, which remains bleached (pale red forespore). Arrows show diffusion of FM4-64 to the forespore. (**B**) Corrected fluorescence recovery (cFR) of individual sporangia without degradation (JLG808) or after degradation in the forespore (F, JLG821), mother cell (MC, JLG823) or both (F and MC, JLG825). When calculating the average cFR (black lines) we excluded curves (pale lines) that resembled those without degradation because they likely represent sporangia in which SpoIIIE is not yet degraded. (**C**) Average cFR in sporangia without SpoIIIE degradation (green, n = 27) or degradation in the mother cell (blue, n = 35), the forespore (orange, n = 33) or both (red, n = 33). Error bars, standard deviation.

The following figure supplements are available for figure 6:

**Figure supplement 1.** Cell-specific degradation of SpoIIIE$^{ATP-}$ mutant.

**Figure supplement 2.** Fluorescence recovery after photobleaching (FRAP) of forespore membranes.

---

ultimately translocate the chromosome in the reverse direction. As expected, degradation in both cells blocked chromosome translocation (normalized DAPI intensity, 21%) and generated 6% anucleate forespores, less than when SpoIIIE was degraded only in the mother cell (*Figure 7B*). Expresison of *sspB$^{Ec}$* in strains in which SpoIIIE was not tagged with SsrA* did not show any defect in chromosome translocation, indicating that the observed phenotypes are indeed due to the degradation of SpoIIIE molecules (*Figure 7—figure supplement 2*). These results support the notion that SpoIIIE functions as a DNA exporter (*Sharp and Pogliano, 2002*), and that although it normally operates in the mother cell, it is capable of operating in the forespore in the absence of the mother cell subcomplex (*Figure 7C*), generating anucleate forespores. Surprisingly, our results also provide evidence that the forespore subcomplex is necessary for maximal DNA translocation efficiency, a result confirmed by quantifying the frequency with which a CFP reporter of forespore gene expression that was positioned near the terminus entered the forespore (*Figure 7—figure supplement 3*).

## Discussion

We investigate here the organization and function of the SpoIIIE DNA translocation channel during *B. subtilis* sporulation using cell-specific protein degradation, quantitative super-resolution microscopy, and FRAP. We were able to directly visualize the assembly of SpoIIIE into two subcomplexes consisting of subunits in each daughter cell. Cell-specific protein degradation and qPALM were used to determine the relative abundance of SpoIIIE in each daughter cell, and our results indicate that each contain approximately equal numbers of SpoIIIE molecules. These observations support a model in which each chromosome arm is transported through a paired channel that spans two lipid

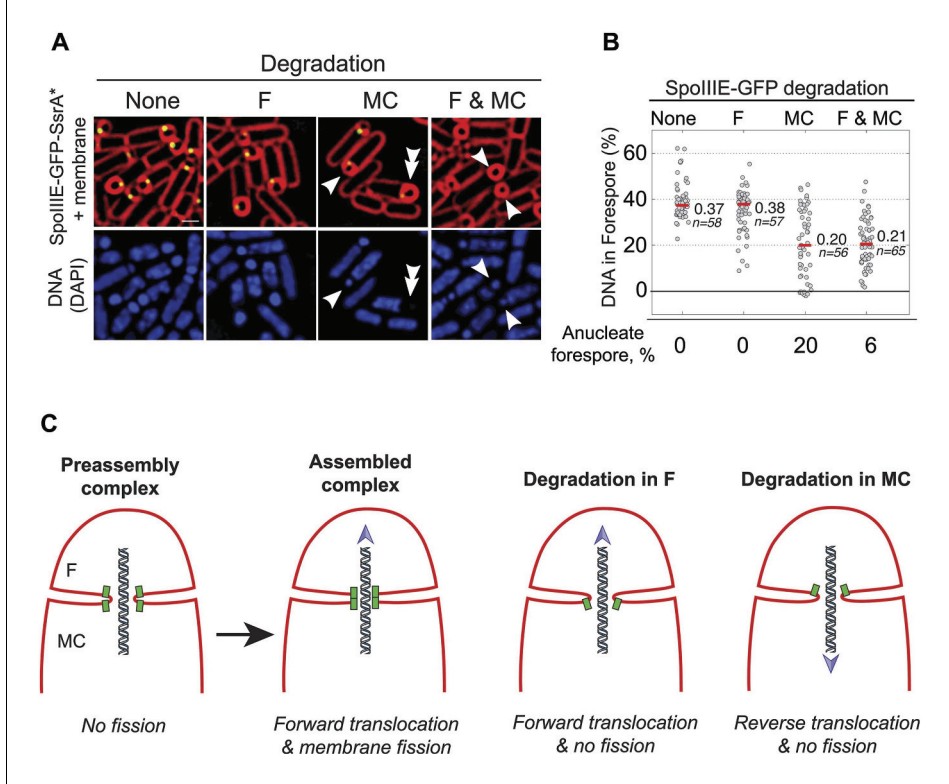

**Figure 7.** Role of forespore and mother cell SpoIIIE in chromssome translocation. (**A**) Visualization of DAPI-stained DNA (blue) after cell-specific degradation of SpoIIIE-GFP-SsrA* (green). Membranes are stained with FM4-64 (red). Single arrowheads show forespores containing small amounts of DNA; double arrowhead shows anucleate forespores. Scale bar, 1 μm. (**B**) Forespore DAPI intensity normalized to the total intensity (forespore plus mother cell) of sporangia about to complete engulfment in the indicated degradation strains. Each dot shows the normalized DAPI intensity of one forespore. Median (red line) and the number of sporangia (n) are indicated. Sporangia containing <0.05 DNA in the forespore were considered anucleate. (**C**) Models for SpoIIIE and membrane organization before and after cell-specific SpoIIIE degradation. During division, SpoIIIE assembles channels that separate daughter cell membranes and mediate forward chromosome translocation. Selective degradation of SpoIIIE either in the mother cell or the forespore reverses membrane fission, with the remaining molecules exporting DNA from their respective cell.

The following figure supplements are available for figure 7:

**Figure supplement 1.** Chromosome translocation at different stages of engulfment.

**Figure supplement 2.** Effect of $sspB^{Ec}$ expression on chromosome translocation.

**Figure supplement 3.** Alternative approach to measure chromosome translocation upon cell-specific degradation of SpoIIIE.

bilayers (*Figure 8B*, *Liu et al., 2006*; *Burton et al., 2007*; *Fleming et al., 2010*). The distance between the clusters in each cell was found to be 55 nm, which is larger than the 40 nm width of the septum in the strains used here, providing some support for a channel architecture in which the motor domains of each complex are within the cytoplasm of the respective daughter cell (*Figure 8B*, left) rather than within the septum (*Figure 8B*, right), although further experiments are required to address this point.

The stability of the SpoIIIE complex during DNA translocation (*Fleming et al., 2010*) allowed us to use qPALM (*Lee et al., 2012*) to image and quantify SpoIIIE in living cells. However, it is important to note that imaging live rather than fixed cells would likely overestimate the dimensions of the complex if SpoIIIE moves during DNA translocation, which could explain why the measured width of

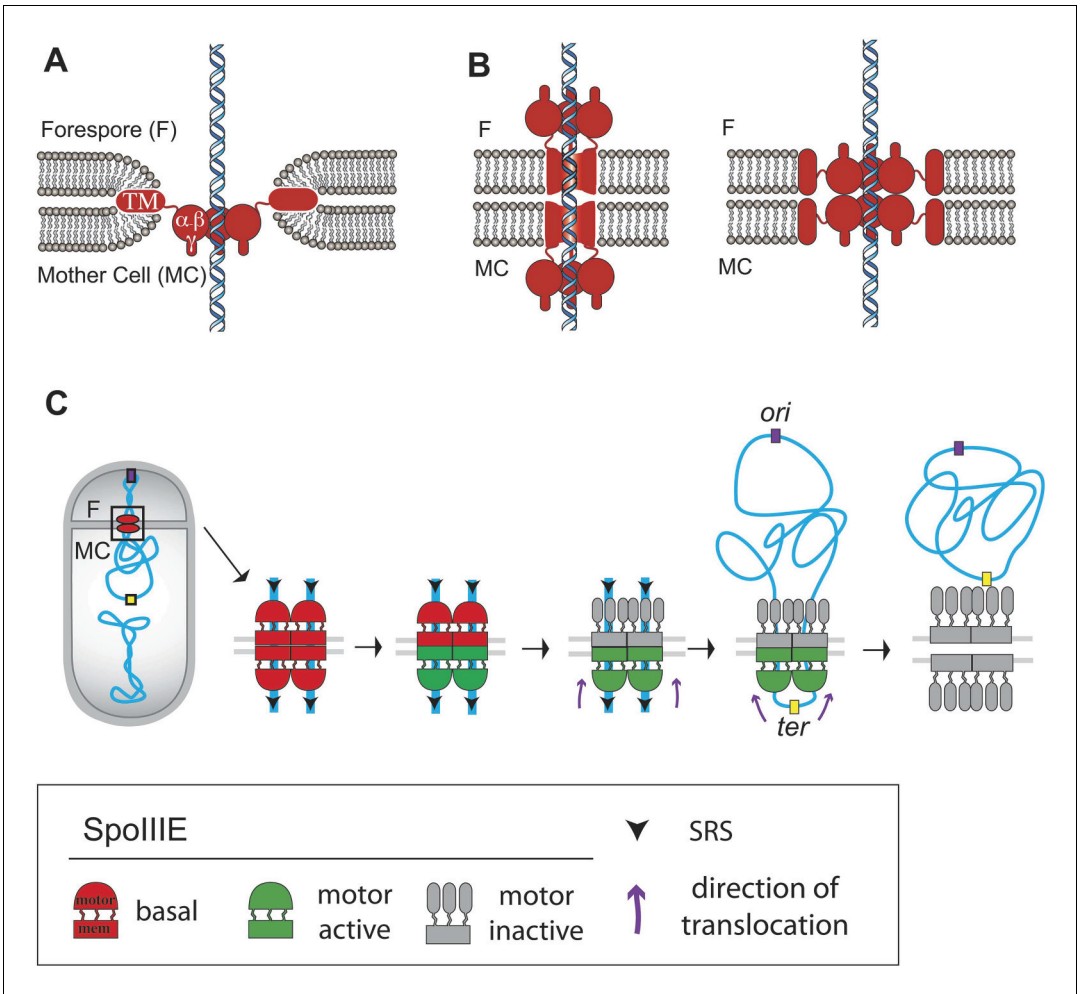

**Figure 8.** Architecture of the SpoIIIE complex during DNA translocation. (**A** and **B**) Septal cross sections showing SpoIIIE (red), DNA (blue) and lipids (grey) and displaying just one DNA strand traversing the septum for simplicity, although at least two arms of the circular chromosome must cross the septum. Our results indicate that the translocation channel has approximately equal numbers of SpoIIIE motor domains in each cell, consistent with the models shown in **B**, but not with the simple aqueous channel model in **A**. They further indicate that the SpoIIIE membrane domains are arranged in such a way that they form a continuous protein barrier at the septum that blocks diffusion within the lipid bilayer and that molecules in each cell are required to form this barrier. We envision two alternative organizations to achieve this. First, the transmembrane domains might form channels in each septal membrane that interact within the septal space and conduct the DNA across the septum, with the motor domains projecting into the cytoplasm (left). Second, the transmembrane domains might form a contiguous external ring that blocks lipid diffusion, with motor domains projecting towards the lumen of an aqueous channel (right). The distance between the centers of the forespore and mother cell clusters (55 nm) is larger than the thickness of the septum (40 nm, *Tocheva et al., 2013*) leading us to prefer the paired channel model (left). (**C**) Each chromosome arm is translocated by a different SpoIIIE channel that is comprised of opposing motor domains that have the ability to export DNA from their respective cell. Translocation directionality is established by the interaction between SRS and the SpoIIIE γ domain, which activates the mother cell motor (green), and inactivates the forespore motor (grey). Consequently, the mother cell-SpoIIIE exports DNA until the final loop (*ter*) reaches the septum. We propose that translocation of the loop generates tension that destabilizes and opens the channels forming a larger pore to pass the loop.

The following figure supplements are available for figure 8:

**Figure supplement 1.** Statistical analysis of qPALM data.

**Figure supplement 2.** Model for chromosome translocation if four DNA strands cross the septum.

the foci (80 nm) is wider than expected for two side by side hexamers (~24 nm). Furthermore, the spatial resolution of PALM (~25 nm) remains significantly larger than the size of most macromolecular assemblies, and it is larger than the distance between the two faces of the septum in wild type cells (~20 nm). We partially overcame the current limitations of PALM by using two genetic tools, the thick septum mutant which increased the distance between the subcomplexes in each cell, and cell-specific protein degradation which improved our ability to discriminate between molecules in each cell.

The method to extract absolute numbers using PALM may depend on other factors, such as label efficiency and inactivation (*Lee et al., 2012*; *Puchner et al., 2013*; *Durisic et al., 2014*). Although our qPALM approach balances the over- and undercounting errors (*Lee et al., 2012*), the numbers of SpoIIIE molecules determined here may represent a lower bound to their true number if some of the photoactivatable fusion domains did not fold properly at the foci. Nevertheless, it is tantalizing to note that our counting method (*Lee et al., 2012*) indicated that the SpoIIIE complex contains 31 (±18) SpoIIIE molecules, enough to form at least two dodecameric channels, one for each chromosome arm (24 molecules in total, *Table 1*). Furthermore, a more detailed analysis of qPALM data revealed the existence of two populations of sporangia with different numbers of SpoIIIE monomers at the septum (~25 and ~50; *Figure 8—figure supplement 1*), suggesting that some translocation complexes have enough molecules to assemble two channels while others could assemble four (*Figure 8—figure supplement 2*). Therefore, we speculate that SpoIIIE organizes around an even number of DNA arms (two or four) during sporulation (*Figure 8—figure supplement 2*), as would be expected if a circular chromosome crossed the septum one or two times, respectively.

Cell-specific protein degradation was used both to quantify and to functionally dissect the role of SpoIIIE molecules in each cell. Our results demonstrate that SpoIIIE is required in both cells to maintain separate septal membranes during DNA translocation and that septal membrane fission is reversible in the presence of trapped DNA (*Figure 6*). The simplest explanation for this finding is that transmembrane domains in each cell assemble a structure that excludes lipids and blocks membrane diffusion between the two cells. This scenario could occur if during assembly of a translocation complex, SpoIIIE membrane domains in the mother cell and the forespore dock via their extracellular domains and the membrane domains of each subunit—within each bilayer—compact into their hexameric form, excluding lipids to form a paired central channel (*Almers, 2001*; *Peters et al., 2001*; *Liu et al., 2006*; *Fleming et al., 2010*). It is possible that docking is indirect and requires additional, accessory proteins that connect the extracellular domains of SpoIIIE. In addition to the data presented here, the paired channel model for septal membrane fission is supported by our previous demonstration that septal membrane fission depends on the SpoIIIE transmembrane domain (*Sharp and Pogliano, 2003*) and its large extracellular loop (*Liu et al., 2006*) and on the assembly of a compact focus (*Fleming et al., 2010*).

In contrast to membrane fission, which requires SpoIIIE in both cells, only mother cell SpoIIIE is essential for DNA segregation, demonstrating that SpoIIIE functions as a DNA exporter (*Sharp and Pogliano, 2002*). While the assembly of SpoIIIE in both sides of the septum provides a structural scaffold for membrane fission, it poses a topological challenge for directional chromosome translocation. Prior to the establishment of directional DNA translocation, both complexes could, in principle, export DNA out of its respective compartment (*Sharp and Pogliano, 2002*; *Becker and Pogliano, 2007*; *Ptacin et al., 2008*) (*Figure 7A,B*), leading to a potential clash between complexes in each cell; as a result, these complexes would compete to translocate DNA either out of or into the forespore. However, our cell-specific degradation data show that chromosome translocation happens most efficiently when both complexes are present (*Figure 7B*, *Figure 7—figure supplement 3*), suggesting that rather than a competition, there is a potentiation between the two hexamers of a channel. We therefore propose that the hexamers within a dodecameric channel should be considered as an integral entity (*Figure 8C*). In this model, docking of the transmembrane domains of hexamers in opposite septal membranes might contribute to the robustness of chromosome translocation in two ways. First, it might stabilize the translocation complex, distributing the tension produced by the movement of the chromosome between the two septal membranes and insulating the DNA from interactions with the membrane or periplasm. Second, it might facilitate the deactivation of the forespore motor domains after activation of the mother cell motor domains by encountering SRSs in the permissive orientation (*Besprozvannaya et al., 2013*; *Cattoni et al., 2013*).

Some members of the SpoIIIE/FtsK/HerA superfamily, including SpoIIIE and FtsK in bacteria (*Burton and Dubnau, 2010*), and possibly HerA in archaea (*Iyer et al., 2004*), are involved in the post-septational segregation of chromosomes when one of them is accidentally trapped in the septum during vegetative cell division. Since the trapped chromosome can belong to either daughter cell, a paired channel similar to the one described here could function as a bidirectional translocation machinery guided by SRS-like sequences that would allow the transport of the chromosome to the appropriate compartment. So, by analogy to SpoIIIE, we hypothesize that the transmembrane domain of these proteins might also form a paired channel crossing both septal membranes.

## Materials and methods

### Strains and cell culture

All the strains used in this study are derivatives of *B. subtilis* PY79. The strains, plasmids and oligonucleotides used in this study, can be found in *Supplementary file 1A–C*. The amino acid sequence of the linker between GFP-SsrA* and the target protein is provided in *Supplementary file 1D*. All the SpoIIIE fusion proteins used here support sporulation with wild type efficiency (*Supplementary file 1E*). Sporulation was induced by resuspension (*Sterlini and Mandelstam, 1969*) at 37°C, except the bacteria were grown in 20% LB prior to resuspension (*Becker and Pogliano, 2007*).

### Plasmid construction

#### pTF25

*dendra2* coding sequence was amplified with primers TF-55 and TF-56. The PCR fragment was digested with SpeI and cloned in pEB410 (*Fleming et al., 2010*) digested with the same enzyme.

#### pJLG3

*ssrA*Ω*kan* fragment was created by amplifying the kanamycin resistant gene of pEB19 (*Becker et al., 2006*) with primers JLG-1 and JLG-5. The fragment was digested with NheI and PstI and was cloned in pBR329 digested with the same enzymes.

#### pJLG7

*sspB* was amplified from genomic DNA of *E. coli* K12 MG1655 with primers JLG-16 and JLG-17, digested with EcoRI and BglII, and cloned in pDG1731 (*Guérout-Fleury et al., 1996*) digested with EcoRI and BamHI.

#### pJLG13

*sspB* gene was amplified with primers JLG-16 and JLG-32 from genomic DNA of *E. coli* K12 MG1655, digested with BamHI and EagI, and inserted in pMDS13 (*Sharp and Pogliano, 2002*) digested with the same enzymes. This plasmid contanis *sspB* under the control of *spoIIQ* promoter and can be integrated into *amyE* locus in *B. subtilis* chromosome.

#### pJLG20

pMDS14 (*Sharp and Pogliano, 2002*) was digested with EcoRI and BamHI, and the resulting fragment containing *spoIID* promoter was cloned in EcoRI-BamHI sites of pJLG7. This plasmid contains *sspB* under the control of *spoIID* promoter and can be integrated into *thrC* locus in *B. subtilis* chromosome.

#### pJLG36

*sfGFP* coding sequence was amplified with primers JLG-33 and JLG-34, digested with SphI and SpeI, and cloned in pJLG3 digested with SphI and NheI. It contains *sfGFP* fused to the *ssrA**, and eight codons encoding an Ala-Ser linker immediately upstream of *sfGFP*.

### pJLG38

pJLG36 was amplified with primers JLG-86 and JLG-87 (both phosphorylated at the 5′end) and religated. The resulting primer contained *sfGFP* coding sequence preceded by eight codons encoding an Ala-Ser linker, without the *ssrA\**.

### pJLG49

This plasmid was constructed by assembling the following four fragments by Gibson Assembly (New England Bioloabs, Ipswich, Massachusetts): (i) the last 852 bp of *sigA* coding sequence (not including the stop codon) amplified with primers JLG-132 and JLG-133 from genomic DNA of *B. subtilis* PY79; (ii) *sfGFP-ssrA\*Ωkan* fragment amplified with primers JLG-7 and JLG-77 from pJLG36; (iii) a fragment of 940 bp corresponding to the region immediately dowstream of *sigA* stop codon, amplified with primers JLG-130 and JLG-131 from genomic DNA of *B. subtilis* PY79; and (iv) a DNA fragment encompassing the spectinomycin resistant gene, the origin of replication, and the ampicillin resistant gene from pDG1662 (*Guérout-Fleury et al., 1996*), amplified with primers JLG-95 and JLG-96.

### pJLG72

This plasmid was constructed by assembling the following four fragments by Gibson Assembly (New England Bioloabs): (i) the last 852 bp of *spoIIIE* coding sequence (not including the stop codon) amplified with primers JLG-234 and JLG-236 from genomic DNA of *B. subtilis* PY79; (ii) *sfGFP-ssrA\*Ωkan* fragment amplified with primers JLG-7 and JLG-77 from pJLG36; (iii) a fragment of 844 bp corresponding to the region immediately dowstream of *spoIIIE* stop codon, amplified with primers JLG-232 and JLG-233 from genomic DNA of *B. subtilis* PY79; and (iv) a DNA fragment encompassing the spectinomycin resistant gene, the origin of replication, and the ampicillin resistant gene from pDG1662 (*Guérout-Fleury et al., 1996*), amplified with primers JLG-95 and JLG-96.

### pJLG112

This plasmid was constructed by assembling the following four fragments by Gibson Assembly (New England Bioloabs): (i) the last 856 bp of *gyrA* coding sequence (not including the stop codon) amplified with primers JLG-416 and JLG-417 from genomic DNA of *B. subtilis* PY79; (ii) *ssrA\*Ωkan* fragment amplified with primers JLG-7 and JLG-184 from pJLG3; (iii) a fragment of 861 bp corresponding to the region immediately dowstream of *spoIIIE* stop codon, amplified with primers JLG-418 and JLG-419 from genomic DNA of *B. subtilis* PY79; and (iv) a DNA fragment encompassing the spectinomycin resistant gene, the origin of replication, and the ampicillin resistant gene from pDG1662 (*Guérout-Fleury et al., 1996*), amplified with primers JLG-95 and JLG-96.

### pJLG118

A 2212 bp DNA fragment of *spoIIIE*^{ATP−} coding sequence was amplified from genomic DNA of KP541 with primers JLG-248 and JLG-450, and assembled with pJLG72 amplified with primers JLG-95 and JLG-245 by Gibson assembly.

### pJLG125

sfGFP was amplified with primers JLG-55 and JLG-77, from pJLG38 and assembled with pJLG112 amplified with primers JLG-539 and JLG-540 by Gibson assembly.

## PALM imaging

Cells were sporulated in presence of 25 ng/ml membrane dye FM5-95 (Life Technologies, Waltham, Massachusetts) and harvested 1 hr and 45 min ($t_{1.75}$) after resuspension. 5 µl of cell suspension mixed with gold fiducial particles was spotted in a poly-L-Lys (Sigma–Aldrich, St. Louis, Missouri) coated coverslip and covered with a second coverslip to generate a cell monolayer (*Fleming et al., 2010*). To minimize the over- and under- quantification of PA-FP molecules: first, the sample was illuminated with the excitation laser (561 nm, 22 mW/mm$^2$) simultaneously with the activation laser (405 nm) whose power varied (from 0 to 9.1 mW/mm$^2$) in time according to the Fermi activation scheme (*Lee et al., 2012*) with parameters tF = 3.2 min and T = 10 s; and second, we employed our previously developed optimal $t_c$ method (*Lee et al., 2012*). PALM data was taken and analyzed using our

custom built microscope and custom written Matlab program (*Fleming et al., 2010*; *Lee et al., 2012*). Single molecule counting was performed in strains with Dendra2 fusion protein.

## Identification and characterization of SpoIIIE clusters in PALM images

### Identification of SpoIIIE clusters (foci)

We first visually inspected each sporangium from the overlaid SpoIIIE-PALM image and membrane labeled image. Then, SpoIIIE-PALM foci were identified as the spatially localized intensely emitting sources in the PALM images that typically organized symmetrically at the mid point of the septum.

### Identification of dual-foci

Only those images in which the two foci could be discerned clearly were identified as dual foci and included in the analysis. Thus, even though in some cases the clusters could be seen as distinct, they were not included in our analysis if they appeared physically connected in the PALM image. To formalize the identification, we employed the Rayleigh resolution criterion. We used the fact that the average width (four times the standard deviation) in perpendicular direction to the septum of the forespore- and mother cell-proximal cluster from dual-foci was 57 nm and 52 nm, respectively (*Figure 5C*). Accordingly, dual foci were identified as such if the distance between the peaks of the individual foci were at least 39 nm or more. It should be pointed out that 94% of the measured distance between these adjacent two clusters are larger than the Rayleigh criteria as shown in *Figure 5A*.

### Characterization of foci

To obtain the position of the clusters we first manually drew a Region of Interest (ROI) that surrounds each cluster and that contains 99% of the fluorescence localization signal. Then, the FM-labeled membrane image was used to draw a line that coincided with the septum across each sporangium (septal line). From the distribution of point localization data of each cluster we obtained the center of mass of that cluster and its dispersion in the direction perpendicular and parallel to the septal line. Four times the standard deviation of these dispersions was used to define the widths of the cluster in each direction.

## Steps for qPALM

Quantitative single-molecule-counting PALM is based on the previously established method (*Lee et al., 2012*) and is carried out through the following steps: (1) obtain the photoactivation rate, blinking rate, and photobleaching rate of a PA-FP that will be used as a fluorescent marker, for instance, through in vitro single-molecule characterization, (2) acquire PALM raw movie data using Fermi-photoactivation scheme, which assures photoactivation of all the PA-FP molecules within a given experimental time at the almost constant rate (*Figure 5—figure supplement 2*), (3) analyze the raw data to localize the appearance of fluorescence bursts in space and time and render a reconstructed PALM image, (4) select a ROI, for example a focus of SpoIIIE, from the visual inspection of the PALM image, and (5) apply the optimal-$\tau_c$ counting (Matlab code available in *Source code 1* also at https://github.com/shyuklee/pafpcluster) to the ROI to obtain the blinking-corrected number of molecules inside the ROI.

## Detailed description of qPALM methodology

The in vitro photophysics of Dendra2 was well described by four states: Nonactive (N), Active (A), Dark (D), and photoBleach (B); and the rates: N to A ($k_a$), A to B ($k_b$), and A to D ($k_d$). The fluorescence recovery, D to A, was well explained by two different rates ($k_{r1}$ and $k_{r2}$) with their population ratio ($\alpha$), which implies the existence of at least two dark states. The photoactivation rate $k_a$ can be externally changed by the intensity of UV light whereas all the other rates are constants fixed by the intrinsic molecular properties of Dendra2. In order to optimize the temporal separation of molecules we changed $k_a$ in time such that a Dendra2 molecule gets photoactivated during a given experimental time, $t_F$, with an almost uniform probability distribution. To account for the overcounting error induced by multiple transitions between the state A and the state D, we introduced a tolerance time of the dwell in the dark states, $\tau_c$: if the fluorescence at the same location recovers within a given $\tau_c$ then the two events are considered due to a blink of an identical Dendra2 molecule rather than due

to two independent photoactivation events of two different molecules. However, the introduction of $\tau_c$ also induces molecular undercounting error because it is probabilistically not impossible that two different Dendra2 molecules actually get photoactivated one after the other within a time interval shorter than the given value of $\tau_c$.

We can achieve the unbiased molecular counting by balancing the overcounting and undercounting through a careful selection of the value of $\tau_c$. This optimal $\tau_c$ depends not only on all the rate constants but also on the actual number of Dendra2 molecules ($N_{mol}$) that we aim at counting by PALM. In the previous work, we obtained the optimal $\tau_c$ by stochastic simulations, but this method was computationally too costly to explore the extensive parameter space defined by the rate constants and $N_{mol}$ (*Lee et al., 2012*). We instead developed an approximate analytic solution of the optimal $\tau_c$ for instantaneous calculation.

If we assume that $N_{mol}$ molecules are independently photoactivated uniformly in time between 0 and $t_F$, then the probability density function of the time lag, $\Delta t_i$, between the two successive $i^{th}$ and $(i + 1)^{th}$ photoactivation events is given by the well known order statistics of the uniform distribution:

$$p(\Delta t_i) = N_{mol} \ (1 - \Delta t_i / t_F)^{N_{mol}}. \tag{1}$$

In order to estimate the undercounting error, we apply a rule of counting for a given $\tau_c$ such that $i^{th}$ and $(i + 1)^{th}$ photoactivations are combined together to be counted as one molecule if $\Delta t_i < \tau_c$. Then the mean of undercounted number, denoted by $\langle N_u \rangle$, can be calculated as follows:

$$\langle N_u \rangle = N_{mol} - \left( 1 + \sum_{i=1}^{N_{mol}-1} \int_0^{\tau_c} p(\Delta t_i) \ d\Delta t_i \right), \tag{2}$$

$$= (N_{mol} - 1)[1 - (1 - \tau_c / t_F)^{N_{mol}}]. \tag{3}$$

The overcounting due to blinking can be estimated using the geometrically distributed probability of the number of transitions that a single Dendra2 molecule makes between the state A and the state D with the dwell in the state D being longer than $\tau_c$ before photobleaching (*Lee et al., 2012*):

$$P\{N_{blink}^{(\tau_c)} = n\} = \overline{\eta} n(1 - \overline{\eta}), \tag{4}$$

where

$$\overline{\eta} = \frac{p_{\tau_c} \eta}{1 - \eta + p_{\tau_c} \eta} \ \text{and} \ p_{\tau_c} = \frac{e^{-k_{r1}\tau_c} + \alpha e^{-k_{r2}\tau_c}}{1 + \alpha}. \tag{5}$$

Then the mean of total overcounted number of independently blinking $N_{mol}$ molecules, denoted by $\langle N_o \rangle$, is given by

$$\langle N_o \rangle = N_{mol} \sum_{n=0}^{\infty} P\{N_{blink}^{(\tau_c)} = n\}, \tag{6}$$

$$= N_{mol} \ \frac{\overline{\eta}}{1 - \overline{\eta}}. \tag{7}$$

Note that $\langle N_u \rangle$ and $\langle N_o \rangle$ are not the exact solutions but just approximate estimates of the undercounting and overcounting because they account for only the situation when the $i^{th}$ photoactivated molecule photobleaches much earlier than the next photoactivation of another molecule. Nevertheless, they are the dominant contributions to the counting error and we can expect to obtain the optimal $\tau_c$ rather accurately from the balance condition, $\langle N_u \rangle = \langle N_o \rangle$, which results in an analytic equation for the optimal $\tau_c$:

$$h(\tau_c; \ N_{mol}, t_F, k_d, k_b, k_{r1}, k_{r2}, \alpha) = 0, \tag{8}$$

where

$$h(\tau_c; \ N_{mol}, t_F, k_d, k_b, k_{r1}, k_{r2}, \alpha) \equiv 1 - (1 - \tau_c / t_F)^{N_{mol}} - p_{\tau_c} \frac{k_d}{k_b} \frac{N_{mol}}{N_{mol} - 1}. \tag{9}$$

This approximate analytic solution agrees well with the result from the stochastic simulation over experimentally realistic range of $N_{mol}$ and $t_F$ when the rates are fixed to the rates of Dendra2 obtained from in vitro single-molecule experiments (*Figure 5—figure supplement 2*).

To count the number of SpoIIIE-Dendra2 molecules inside a cluster, we iterated the feedback between $\tau_c$ and $N_{mol}$ using the in vitro Dendra2 kinetic rates and the analytic solution of optimal $\tau_c(N_{mol})$ until convergence (*Lee et al., 2012*).

## Statistical analysis of qPALM data

### Gaussian Kernel Density Estimator (GKDE)

GKDE is independent of bin size, and therefore a more accurate estimator than a histogram of an unknown probability distribution. It is commonly used across a broad range of disciplines, including single-molecule studies. We used a built-in Matlab function 'ksdensity' that implemented a GKDE algorithm provided in *Botev et al. (2010)*. We first calculated a GKDE curve form the data, and then fitted it to either a single Gaussian or a mixture of two Gaussian functions.

### Variational Bayesian Gaussian Mixture Model (VBGMM)

To further justify the existence of two populations in the SpoIIIE counting data (*Figure 8—figure supplement 1*), we employed VBGMM. Basically, Gaussian Mixture Models (GMM) treats the data by a mixture of multiple Gaussian components. Then, one can fit the GMM model distribution directly to the data without resorting to a histogram or GKDE curve using various methods; among these the simplest is the maximum likelihood method. However, the downside of the maximum likelihood method is that it cannot uniquely determine the number of Gaussian components—denoted by 'K' henceforth—due to the problem of overfitting data. Therefore, the maximum likelihood method is not ideal because it relies on a manual input of 'K' that is predetermined in a subjective and heuristic manner. In contrast, the VBGMM is a Bayesian approach in which the model parameters, such as the mean and the standard deviation of a Gaussian component, are regarded as random variables. The probability distribution of the model parameters, defined as a 'posterior' distribution, is then marginalized and optimized to determine the 'K' value in the VBGMM method (*Bishop, 2006*). VBGMM determines the optimal 'K' that best explains the data without the problem of overfitting. VBGMM automatically takes care of the problem of overfitting within its self-contained statistical framework. This Bayesian approach has been proved to be very successful in objectively determining the number of states from single-molecule FRET data (*Bronson et al., 2009*).

## Fluorescence deconvolution microscopy: imaging and analysis

Samples from sporulating cell cultures were taken 2.5 hr after resuspension ($t_{2.5}$) and added to agarose pads supplemented with 0.5 µg/ml of FM4-64 (Life Technologies) for membrane visualization and 40 ng/ml of DAPI (Invitrogen, Waltham, Massachusetts) for DNA visualization. Images were collected using an Applied Precision optical sectioning microscope equipped with a Photometrics CoolsnapHQ[2] camera using identical exposure times for each sample. Images were deconvolved and analyzed with SoftWoRx version 5.5 (Applied Precision, Issaquah, Washington). Sporangia with flat, curved, and engulfing septa (as defined in *Sharp and Pogliano, 1999*) from 2–4 microscopy fields were quantified. To determine SpoIIIE-GFP-SsrA* foci intensities, GFP intensities of eight optical sections covering a total thickness of 1.05 µm within the cells were summed using SoftWoRx Z-projection tool. The intensity of each projected GFP focus was determined by drawing a ~150 nm$^2$ circumference centered in the middle of the focus, subtracting the average background intensity for each field. The average focus intensity after SpoIIIE-GFP degradation in the mother cell or in the forespore was normalized to the average intensity when SpoIIIE-GFP was not degraded.

To determine the fluorescence intensities of GyrA-GFP-SsrA* and SigA-GFP-SsrA* in the forespore and mother cell, pixel intensities of four optical sections from deconvolved images covering a total thickness of 0.45 µm were summed. Mean GFP intensities of the forespore and the mother cell were determined separately by drawing a polygon encompassing the whole area of every cell. The ratios were calculated after subtracting the mean background intensity. For each graph, values were made relative to the average ratio of cells with flat septum in the non-degradation strain.

To determine the degree of DNA translocation, DAPI pixel intensities of four optical sections covering a total thickness of 0.45 µm were summed. Samples were taken 3 hr after resuspension ($t_3$), and

DAPI intensities of forespore (F) and mother cell (MC) from sporangia about to complete engulfment were determined separately by drawing a polygon encompassing the whole DNA area of every compartment. After subtraction of the average background intensity, the normalized DAPI intensity [F intensity/(F intensity + MC intensity)] was determined for each sporangium.

### FRAP

FRAP of forespore membranes was performed as described (*Fleming et al., 2010*). Briefly, cells were sporulated in the presence of 2 µg/ml FM4-64. After 2.5 hr, cells were washed three times with sporulation medium and placed onto 1.2% agarose pads. Forespore membranes were bleached with 0.3 s pulse from a 488-nm argon laser set to 30% power, and membrane images collected at appropriate time intervals. FRAP quantification was performed as described (*Fleming et al., 2010*).

## Acknowledgements

We thank Joerg Schnitzbauer for helping develop the algorithm to characterize SpoIIIE foci from PALM data, Gheorghe Chistol for critical reading of the manuscript and Ninning Liu for helpful discussion. JLG was funded by an EMBO long-term postdoctoral fellowship. The work in the Bustamante lab was supported in part by grants from the National Institutes of Health (GM032543), the U.S. Department of Energy, Office of Basic Energy Sciences, Division of Materials Sciences and Engineering (Contract no. DE-AC02-05CH11231) and the Howard Hughes Medical Institute. The work in the Pogliano lab was supported by the National Institutes of Health (GM57045).

## Additional information

### Funding

| Funder | Grant reference number | Author |
|---|---|---|
| National Institutes of Health | GM57045 | Kit Pogliano |
| U.S. Department of Energy | DE-AC02-05CH11231 | Carlos Bustamante |
| Howard Hughes Medical Institute | | Carlos Bustamante |
| EMBO | long-term postdoctoral fellowship | Javier Lopez-Garrido |
| National Institutes of Health | GM032543 | Carlos Bustamante |

The funders had no role in study design, data collection and interpretation, or the decision to submit the work for publication.

### Author contributions

JYS, Constructed the strains to visualize SpoIIIE by PALM, Performed the PALM experiments, Analysed the PALM data, Developed the project and interpreted the results, Wrote the manuscript; JL-G, Developed the cell-specific protein degradation system and performed the experiments and analyzed the data of FRAP and chromosome translocation efficiency, Constructed the strains to visualize SpoIIIE by PALM, Developed the project and interpreted the results, Wrote the manuscript; S-HL, Developed the algorithm for qPALM, Developed the project and interpreted the results; CD-C, Performed the PALM experiments; TF, Constructed the strains to visualize SpoIIIE by PALM; CB, KP, Developed the project and interpreted the results, Wrote the manuscript

## Additional files

### Supplementary files

• Supplementary file 1. Strains and plasmids. (A) Strains used in this study. (B) Plasmid used in this study. (C) Oligonucleotides used in this sudy. (D) Sequence of amino acid residues of the linkers within the SpoIIIE-GFP-SsrA*. (E) Spore titers of strains containing the different *spoIIIE* fusion proteins used in this study.

• Source code 1. qPALM analysis MATLAB package.

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
