## [Decision Letter]

Thank you for sending your work entitled “Visualization and functional dissection of coaxial paired SpoIIIE channels across the sporulation septum” for consideration at *eLife*. Your article has been favorably evaluated by Richard Losick (Senior editor) and two reviewers, one of whom, Xiaowei Zhuang, is a member of our Board of Reviewing Editors.

The Reviewing editor and the other reviewer discussed their comments before we reached this decision, and the Reviewing editor has assembled the following comments to help you prepare a revised submission.

The authors examine the spatial distribution and copy number of the translocase SpoIIIE in the septum of sporulating *B. subtilis* using a combination of elegant genetic approaches, fluorescent imaging, and super-resolution microscopy. The spatial organization of this protein has been the subject of studies over the past few years, and the authors provide convincing data in support of a clear model for the spatial organization of this protein at the septum. They showed that SpoIIIE in the septum are present on both the mother cell side and the forespore side, and both sides are needed for the cell fission. They further showed that the SpoIIIE complex functions to export the DNA out of the cell, and only the SpoIIIE complex on the mother cell side is required for the translocation of the DNA into the forespore. Interestingly, the complexes on the two sides do not compete with each other, but cooperate for efficient DNA segregation. The results presented in this manuscript are novel, important and beautiful. Both the Reviewing editor and the other reviewer recommend the publication of this work in *eLife* with only minor revisions.

The following minor comments are for the authors' consideration to improve their already excellent manuscript. If the authors think some of the suggested experiments or analyses are not doable or unnecessary, please provide reasons in the revision cover letter.

Reviewer #1:

I have only a few minor points. These points only need to be addressed by text additions and figure presentation changes, and there is no need of additional experiment or analysis.

1) It is now known that not 100% of the photoactivatable fluorescent proteins are fluorescent or photoactivatable, but some molecules remain dark. Please discuss how this will affect the quantification of monomer numbers in the SpoIIIE complex.

2) The authors explain that P_*spoIIQ*_ and P_*spoIID*_ are only activated after polar septation and only in a compartment specific fashion. While this property of these promoters is widely known, a reference to this point when these promoters are mentioned might help the general reader.

3) The black bars in Figure 3 and Figure 6—figure supplement 1 showing the average are not very clearly visible.

Reviewer #2:

1) The cell-specific degradation technique is powerful. But the proof-of-concept data shows a few snap shots sparsely operated in time. A more dense time series would be more convincing.

2) What should I make of the 19±11 and 15±8 molecules in the mother cell- and forespore-proximal complexes? Do the authors believe the true number is 12 and 12? What sets the limits of this counting technique? If the authors could lower the error it would be quite powerful.

---

## [Author Response]

Reviewer #1:

*I have only a few minor points. These points only need to be addressed by text additions and figure presentation changes, and there is no need of additional experiment or analysis*.

*1) It is now known that not 100% of the photoactivatable fluorescent proteins are fluorescent or photoactivatable, but some molecules remain dark. Please discuss how this will affect the quantification of monomer numbers in the SpoIIIE complex*.

Even though our qPALM approach balances the over- and undercounting errors generated due to the blinking of PA-FPs (Lee et al., 2012), it is correct that less than 100% labeling and/or improperly folded PA-FPs can introduce errors in the counting. The label efficiency is much less of an issue in PALM because the labels are genetically encoded, and we used western blots to confirm that 100% of the SpoIIIE molecules were fused to PA-FPs, Dendra2 or tdEOS (Supplementary file 2). However, it is possible to imagine that some of the PA-FP domains may not have folded properly and so may remain dark. At this point in time we do not know any method to correct for this effect or to measure the fraction of unfolded or otherwise non-fluorescent proteins. Accordingly, we have now modified the text as follows in the Discussion section:

“… Although our qPALM approach balances the over- and undercounting errors, (Lee et al, 2012), the numbers of SpoIIIE molecules determined here may represent a lower bound to their true number if some of the photoactivatable fusion domains did not fold properly at the foci. Nevertheless, …”

*2) The authors explain that P*_spoIIQ_*and P*_spoIID_*are only activated after polar septation and only in a compartment specific fashion. While this property of these promoters is widely known, a reference to this point when these promoters are mentioned might help the general reader*.

We have introduced the references below to address this point:

a) For a description of *spoIIQ* expression:

Londoño-Vallejo J, Fréhel C, Stragier P. 1997. *spoIIQ*, a forespore-expressed gene required for engulfment in *Bacillus subtilis*. Molecular Microbiology 24:29-39. doi:10.1046/j.1365-2958.1997.3181680.x.

b) *spoIID* expression was first characterized in the following papers:

Rong S, Rosenkrantz MS, Sonenshein AL. 1986. Transcriptional control of the *Bacillus subtilis spoIID* gene. Journal of Bacteriology 165:771-9.

Clarke S, Lopez-Diaz I, Mandelstam J. 1986. Use of *lacZ* gene fusions to determine the dependence pattern of the sporulation gene *spoIID* in spo mutants of *Bacillus subtilis*. Journal of General Microbiology 132:2987-94. doi:10.1099/00221287-132-11-2987.

c) Microscopy images of sporangia expressing GFP from *spoIIQ* or *spoIID* promoters can be found in Supplemental figure 2 of the following paper:

Sharp MD, Pogliano K. 2002. Role of cell-specific SpoIIIE assembly in polarity of DNA transfer. Science 295:137–9. doi:10.1126/science.1066274.

*3) The black bars in*
Figure 3
*and*
Figure 6—figure supplement 1
*showing the average are not very clearly visible*.

Thank you for pointing this out. We have lightened the background color and changed the style of the black bars so that they are more visible.

Reviewer #2:

*1) The cell-specific degradation technique is powerful. But the proof-of-concept data shows a few snap shots sparsely operated in time. A more dense time series would be more convincing*.

We agree with the reviewer that a more continuous time series would be ideal to study degradation dynamics. We have attempted to address this point by performing time-lapse microscopy of individual sporangia, but our results have been compromised by phototoxicity (to which sporulating cells appear particularly sensitive) and GFP photobleaching. In the absence of solid timelapse data, we were left with the option of classifying sporangia according to the different engulfment stages that can be unambiguously distinguished using membrane dyes. We here used the five stages of engulfment that are used for this purpose (Sharp and Pogliano, 1999), although we did not include the two final stages, since at this point, degradation is complete. The addition of intermediate stages based simply on visual screening would be subjective. Nevertheless, we are continuing to develop the cell-specific degradation system, and a key priority is to improve the temporal resolution.

*2) What should I make of the 19±11 and 15±8 molecules in the mother cell- and forespore-proximal complexes? Do the authors believe the true number is 12 and 12? What sets the limits of this counting technique? If the authors could lower the error it would be quite powerful*.

Two main sources set the limit of the counting technique, the counting error (or uncertainty) and the inherent variation of the biological structure composition. Unfortunately, we do not have any direct method to decouple the different sources of the dispersion in the results. However, here we will try to do a back-of-the-envelope calculation according to our previous work based on the in vitro characterization of Dendra2 photophysics, where we demonstrated that the counting error depended on the number of molecules and the length of the experiment (Figure S8, Lee et al. 2012). We estimate that the molecular counting error contributes 3 and 2 monomers for populations peaked at ∼19 and ∼15, respectively (Figure 5D center and right), which is smaller than the standard deviation of the measured distribution (11 and 8). Then, the remaining standard deviation (after eliminating the contribution of the counting error), 10 and 7, should correspond to the inherent variation of the biological structure. This suggests that most of the variability in the number of monomers is due to biological variation in the stoichiometry of the SpoIIIE complex rather than counting errors, and that it would not be possible to narrow the variance by just lowering the counting uncertainty.

It is very likely that SpoIIIE molecules that are not integrated into a functional channel co-exist with the channels on both subcomplexes. Note that the effect of these extra “free” monomers would be to overestimate the number of monomers in the channel and would tend to compensate the underestimation arising from the improper folding of the PAFP’s described above (response to the Reviewer’s comment 1). Nonetheless, the contribution of these two sources of stochastic variation will necessarily add when estimating the variance of the distribution. Thus, we have more confidence in the first than in the second moment of the distribution.